# Hepatic Transcriptomics Reveals Reduced Lipogenesis in High-Salt Diet Mice

**DOI:** 10.3390/genes14050966

**Published:** 2023-04-24

**Authors:** Jing Xu, Fei Mao, Yan Lu, Tiemin Liu, Xiaoying Li, Yao Li

**Affiliations:** 1Department of Endocrinology and Metabolism, Zhongshan Hospital, Fudan University, Shanghai 200032, China; xujing_272@126.com (J.X.); medichem@126.com (F.M.); 2Institute of Metabolism and Regenerative Medicine, Shanghai Sixth People’s Hospital Affiliated to Shanghai Jiao Tong University School of Medicine, Shanghai 200233, China; rjluyan@126.com; 3School of Life Sciences, Fudan University, Shanghai 200032, China; tiemin_liu@fudan.edu.cn; 4Department of Laboratory Animal Science, Shanghai Jiao Tong University School of Medicine, Shanghai 200032, China

**Keywords:** metabolism, transcriptomics, high-salt diet, liver, fatty acid profile

## Abstract

It has been demonstrated that a high salt diet (HSD) increases the risk of cardiovascular disease and metabolic dysfunction. In particular, the impact and molecular mechanisms of long-term HSD on hepatic metabolism remain largely unknown. To identify differentially expressed genes (DEGs) affecting the metabolism of liver tissues from HSD and control groups, a transcriptome analysis of liver tissues was performed in this study. As a result of the transcriptome analysis, the expression of genes related to lipid and steroid biosynthesis (such as *Fasn*, *Scd1*, and *Cyp7a1*) was significantly reduced in the livers of HSD mice. Additionally, several gene ontology (GO) terms have been identified as associated with metabolic processes in the liver, including the lipid metabolic process (GO: 0006629) and the steroid metabolic process (GO: 0008202). An additional quantitative RT-qPCR analysis was conducted to confirm six down-regulated genes and two up-regulated genes. Our findings provide a theoretical basis for further investigation of HSD-induced metabolic disorders.

## 1. Introduction

Most modern diets include salt as a taste enhancer and preservative, and it is crucial to the health of both animals and humans. In most populations worldwide, the average dietary salt consumption has increased alarmingly, greatly exceeding the recommended consumption by the World Health Organization (WHO) (5 g/d) and has eventually evolved into a prominent dietary concern around the world [1,2]. Additionally, despite various salt-reduction measures implemented over the past two decades, the amount of daily sodium consumed per capita has remained constant [3]. It is well known that excessive consumption of salt can increase the risk of hypertension and existing cardiovascular disease [4,5]. Increasing salt intake also contributes to an increased risk of metabolic disorders like obesity [6,7,8,9], insulin resistance (IR) [10], type 2 diabetes (T2DM) [11,12], metabolic syndrome (MetS) [13,14], and sarcopenia [15]. In light of this, it is meaningful to understand how excessive intake of salt contributes to these health conditions.

As the liver serves a crucial role among many aspects of homeostasis and nutrient metabolism, dietary high salt intake is of particular concern. Accumulated salt in tissue causes osmotic pressure-dependent enrichment of proinflammatory immune cells [16,17]. As the sole organ in the human body capable of providing nutrients for energy production to other tissues, the liver obtains the majority of its blood supply from the intestine via the portal vein (approximately 70%), which is loaded with bacterial products, environmental pollutants, and dietary antigens [18,19]. Upon the hepatocytes being damaged, they could release cytokines and chemokines that promoted inflammatory responses, activating both resident and invading immune cells in the liver [20]. Previous studies have already demonstrated that an increased salt intake has been independently correlated with an elevated risk of several liver-associated diseases with reference to population studies and mice experiments, such as nonalcoholic fatty liver disease (NAFLD) [21,22], progressive liver fibrosis [23,24]. Recently, our study found that HSD may affect the expression and levels of organokines in metabolic tissues, thereby mediating crosstalk across metabolic tissues [25].

Our study is designed to discover the metabolism-relevant gene alterations associated with high salt stress in the liver using RNA-Seq data. Here, we mainly focus on the changes in gene expression involved in de novo lipogenesis and cholesterol biosynthesis in response to long-term salt stress. It provides a metabolic profile under chronic salt stress, which may contribute to a better understanding of the fundamental mechanisms underlying different phenotypes.

## 2. Materials and Methods

### 2.1. Experimental Design and Animals

Female C57BL/6 mice aged 6 weeks were purchased from Lin Chang Laboratory Animal Care, Shanghai, and maintained under specific pathogen-free conditions in the Department of Laboratory Animal Science (Shanghai Jiao Tong University School of Medicine, China). All the female mice were habituated for a period of 1 week for regular chow upon arrival. Then all the mice were divided randomly into 2 groups: a standard diet (0.4% NaCl) and a high-salt diet (8% NaCl and 0.9% saline solution for 3 months). All the female mice were kept at 21 °C [+/−1 °C] with a humidity of 55% [+/−10%] and a 12-h light/dark cycle. Liver tissues were extracted from 3 HSD and 3 control mice, respectively. High-throughput sequencing of the transcriptome of the livers was performed (Appendix A).

### 2.2. Immunohistochemistry

After carefully washing twice with PBS and fixing the liver tissues with 4% polyformaldehyde for 15 min, oil red O staining was performed according to the manufacturer’s instructions using a kit from Jiancheng Biotech (#D027; China).

### 2.3. Biochemical Tests

To measure liver biochemistries, the mice were sacrificed after 6 h of fasting. All the livers were harvested and snap-frozen in liquid N_2_. For Oil Red O staining, liver samples were rapidly fixed, embedded, and cut into 8-μm sections. Hepatic lipids were extracted with chloroform–methanol (2:1), as previously described [26]. The levels of hepatic triglycerides, total cholesterol, and non-esterified fatty acid (NEFA) were detected using commercial kits from Jiancheng Bioengineering Institute (Cat No. A042-2-1, Nanjing, China) according to the manufacturer’s instructions.

### 2.4. RNA Isolation

A total of 5 mg of liver tissue specimens in liquid nitrogen was immersed into 500 mL of TRIzol reagent. To the 500 μL of TRIzol-liver mixture, chloroform was added and then mixed. After 5 min at room temperature, the mixture was then centrifuged at 12,000 rpm for 15 min at 4 °C. The top phase was then isolated and then mixed with 100% ethanol (1:1 volume ratio). All total RNA samples were used for genome-wide mRNA sequencing by Novogene Corp (Sacramento, CA, USA). All RNA samples were inspected for quality with the following steps before the construction of the library: (1) RNA concentration and purity check, OD260/OD280 ratio of 1.8–2.0 (Nanodrop); (2) RNA integrity and DNA contamination (agarose gel electrophoresis); and (3) RNA integrity confirmation (Agilent 2100 Bioanalyzer).

### 2.5. Library Construction and Sequencing

In order to construct RNA libraries, rRNA-depleted RNAs were used along with the TruSeq Stranded Total RNA Library Prep Kit (Illumina, San Diego, CA, USA) as directed by the manufacturer. The BioAnalyzer 2100 (Agilent Technologies, Inc., Santa Clara, CA, USA) was then used for quality control and quantification of libraries. Single-stranded DNA molecules from 10 pM libraries were denatured by Illumina flow cells, amplified in situ as clusters and sequenced on the Illumina HiSeq Sequencer for 150 cycles finally.

### 2.6. Data Processing

Illumina HiSeq 4000 sequencer reads were paired-end and quality controlled by Q30. Amplification of the 3′ adaptor and the removal of low-quality reads were performed by cutadapt software (v1.9.3), followed by alignment with the reference genome (UCSC MM10) using hisat2 software (v2.0.4). Guided by the Ensembl gtf gene annotation file, cuffdiff software was then used to get the gene level fragments per kilobase per million (FPKM) as the expression profiles of mRNA and fold change. Based on FPKM, *p*-values were calculated and differentially expressed mRNAs were identified. Our differentially expressed mRNAs were analyzed with GO enrichment analysis and KEGG (Kyoto Encyclopedia of Genes and Genomes) enrichment analysis. The FPKM values were used to normalize the read counts. The squares of Pearson coefficient r values were calculated to show correlations between samples and reproducibility based on the FPKM of each gene in each sample. The FPKM value of each gene was averaged across groups, and log2(FPKM+1) values were used to generate a heatmap showing how genes and groups clustered. Based on log2(FPKM+1) of all genes, the Euclidean distance method between groups was calculated.

### 2.7. RT-qPCR Validation

Applied Biosystems’ PowerUp SYBR Green Master Mix was used for the synthesis of cDNA strands from purified total RNA. For qPCR, the StepOnePlus kit was used for qPCR. And for quantification, the StepOnePlus real-time PCR system (Applied Biosystems) was used. *Gapdh* expression values were used to normalize qPCR results. All the primer sequences used in this study were listed in Appendix A.

### 2.8. Statistical Analysis

GraphPad 8.0 was used for statistical analyses. Data are expressed as mean ± standard mean of error (SEM). *p* < 0.05 was used to determine the statistical significance of the difference between the 2 groups using Student’s *t*-tests.

## 3. Results

### 3.1. Metabolic Phenotypes of HSD Mice

After 12 weeks of sodium-rich chow, the body weight of HSD mice was substantially decreased in comparison with that of the controls (Figure 1A), while a significant increase was observed in the ratio of liver weight to body weight in HSD mice (Figure 1B). Surprisingly, a remarked decrease in lipid content was seen in HSD mice as compared with the mice fed normal chow using Oil Red O staining (Figure 1C). In parallel, total intrahepatic triglyceride (TG) content was significantly reduced in HSD mice, whereas total cholesterol (TC) and NEFA contents were not significantly different between HSD and normal chow groups (Figure 1D–F). To screen differentially expressed genes in liver tissues of HSD mice and further obtain a global gene expression pattern related to HSD-induced liver metabolic characteristics, a transcriptome analysis was performed in livers from HSD and the control mice. The 6 Gb row data was obtained from each sample, and high-quality sequences with approximately 44–46 Mb reads were obtained from the HSD and the control mice. The mapping rates to relevant genomes were from 96.51% to 97.17% (Appendix A). For a more accurate assessment of gene expression trends, transcript abundances were normalized by fragments per kilobase of exon per million (FPKM) mapped fragments. Data showed that the mRNA-Seq results were both reliable and reproducible. The RNA-Seq dataset was subjected to a principal component analysis (PCA) to provide an overall view of the transcriptomes from two different groups (Appendix A), which indicated that transcriptome results were highly reproducible and reliable.

### 3.2. Analysis of Differentially Expressed Genes

During the identification of DEGs, a log2 fold change (log2|FC|) ≥ 1 and a q-value (adjusted *p*-value) < 0.01 were used as screening criteria throughout the differential analysis. Regarding the liver transcriptome, based on the comparison of the liver transcriptomes of the HSD and the control mice, 1204 DEGs were identified (*p* adjusted value < 0.01) (Appendix A). Of those, as demonstrated in volcano plots of up- and down-regulated genes, 466 genes (38.70%) were highly expressed in HSD mice compared with the control mice and were referred to as “up-regulated,” while the remaining 738 genes (61.30%) had lower levels of expression in HSD mice and were termed “down-regulated” (Figure 2A). And the heatmap in Figure 2B showed the expression patterns between DEGs of two groups. The pie diagram depicts the total number of up- and down-regulated genes in response to salt-induced stress (Appendix A). Overall, the changes were relatively remarkable since the log2 fold changes ranged from −7.34 to 7.51. The genes ranked as the top 20 log2 fold changes (downregulated and upregulated) were presented in Table 1 and Table 2. Transcripts decreased in the HSD mice included genes related to hepatic sulfonation of bile acid, G protein-coupled receptor signaling pathway, immune response, and cAMP-mediated signaling. In addition, there were a number of other genes upregulated in the HSD mice involved in protein polyglycylation, ferritin receptor activity, monooxygenase activity, positive regulation of phagocytosis, insulin receptor signaling pathway, AMP-activated protein kinase activity, glycogen biosynthetic process, glycolytic process. In summary, the significantly differentially expressed genes participating in hepatic glucose and lipid metabolism of HSD mice are listed in Table 3.

### 3.3. GO and KEGG Pathway Analysis of DEGs

We also performed a GO enrichment analysis to explore the potential function of DEGs regarding the features of hepatic metabolic regulation. An enrichment test was applied to search for significantly overrepresented GO terms (*p*-value < 0.01) and KEGG pathways (*p*-value < 0.01). After the GO function analysis, a total of 236 GO entries were obtained (*p* < 0.01), and the top 10 items of biological processes (BP), cellular components (CC), and molecular function (MF) were selected for visualization (Figure 3A–C). As shown in Figure 3A and Appendix A, the GO clusters were highly enriched for metabolic processes, especially lipid metabolism. The top five BP of GO terms were “lipid metabolic process” (GO: 0006629), “steroid metabolic process” (GO: 0008202), “circadian rhythm” (GO: 0007623), “fatty acid metabolic process” (GO: 0006631), and “cholesterol homeostasis” (GO: 0042632). All GO results are shown in Appendix A.

Here, based on the KEGG pathway database, we systematically performed a pathway enrichment analysis of these DEGs. In this study, all 897 DEGs were assigned to 37 KEGG pathways. The top 20 enriched KEGG pathways of the two groups are presented in Figure 3D. Among these, metabolic pathways (map01100) were highly enriched according to KEGG pathways. For HSD treatment, some KEGG pathways were also related to typical signal transduction (such as the PPAR signaling pathway, AMP-activated protein kinase (AMPK) signaling pathway, TGF-beta signaling pathway), chemical carcinogenesis (DNA adducts; receptor activation), retinol metabolism, drug metabolism (cytochrome P450 and other enzymes), circadian rhythm, and cholesterol metabolism (Figure 3D). All the KEGG results are shown in Appendix A. Furthermore, Table 4 and Appendix A showed the information on the top 10 significantly enriched canonical pathways containing DEGs relevant to HSD (*p* < 0.01).

### 3.4. Transcriptome Data Validation by qPCR

To further validate the results observed based on the RNA-seq data, the hepatic gene expression of eight genes was quantified with RT-qPCR in the 6 liver samples, which were selected to represent the lipid metabolic process. Six lipid metabolism-related genes (*Acly*, *Fasn*, *Scd1*, *Cd36*, *Acaca*, and *Acot1*) and two key gluconeogenic genes (*Pck1* and *G6pc*) were validated by qPCR (Figure 4). Additionally, six more DEGs among the up-regulated group and four classical metabolic pathways (PPAR signaling pathway, Retinol metabolism, Bile secretion, and Cholesterol metabolism) were also selected for further validation, which is illustrated in Appendix A, respectively. All the above validation results generally agreed well with that of transcriptome sequencing data.

## 4. Discussion

As the major metabolic organ, the liver plays a central role in maintaining whole-body metabolic homeostasis, including glucose and lipid metabolism. In addition, as an important integrator of nutrient metabolism, the liver is also involved in a variety of key signaling pathways such as insulin receptors, PPARα and mTORC1 signaling. Therefore, a well understanding of the liver metabolism in response to different nutrients is clearly warranted. In this study, there was a reduction in hepatic triglyceride contents by dietary salt overload (Figure 1). To investigate the underlying molecular changes caused by chronic consumption of a high-salt diet, we also had previously assessed several genes involved in de novo lipogenesis and cholesterol biosynthesis [25]. However, the underlying mechanism of reduction in triglycerides induced by chronic salt-loading was still unclear since several genes were involved in this process.

In the case of lipid metabolism, those genes involved in fatty acid synthesis (*Acss2*, *Fasn*, *Acaca*, *Me1*, *Scd1*, *Elovl6*, *Acly*, and *Acc*), fatty acid transport (*Cd36*, *Fabp5*, and *Acsl3*), triglyceride synthesis (*Gpam*), cholesterol synthesis (*Sc5d*), and adrenal steroid (*Cyp17a1*) were down-regulated, whereas genes related to fatty acid oxidation (*Cyp4a10*, *Cyp4a14*, and *Cyp4a31*) and bile acids synthesis (*Cyp7a1*) were significantly up-regulated in response to high-salt stimulation (Table 3).

Of particular concern would be the expression of the lipogenic genes. Our pathway analysis showed that there was a universal decrease in metabolic pathways for hepatic lipid synthesis, including *Fasn*, *Acly*, and *Scd1*. Similarly, down-regulation was also observed in genes related to lipid uptake, such as *Cd36* and *Fabp5*, suggesting that such a high salt could suppress hepatic lipid accumulation through the inhibition of de novo lipogenesis and lipid uptake.

So far, quite a few numbers of pathways have been reported to be activated by high-salt diet challenge, including peroxisome proliferators activated receptor (PPAR) signaling pathway, steroid hormone biosynthesis, AMPK signaling pathway and PI3K-Akt signaling pathway [10,42]. Nevertheless, in our RNA-seq results, we have also identified some novel signaling pathways. To our knowledge, a significant enrichment of the FoxO signaling pathway, glutathione metabolism, fatty acid biosynthesis and insulin resistance was not ever reported in HSD mice previously. Overall, our transcriptomics results suggest that high-salt feeding resulted in significant weight loss and a reduction of hepatic lipid accumulation by suppressing lipogenesis and promoting lipid oxidation through the PPAR pathway (Figure 3). These findings could help further explore the mechanisms for metabolic dysfunction caused by a high-salt diet.

It is noteworthy that the association between high salt consumption and obesity has been previously reported in various reports independent of energy intake [43,44]. In rats, long-term salt overload promoted adipocyte hypertrophy [45,46]. Salt overload may also stimulate lipogenesis in adipocytes and induce inflammatory adipocytokine secretion, explaining sodium-associated obesity’s inflammatory adipogenic process [47]. In contrast, few reports have focused on the metabolic dysfunction occurring in the liver upon consumption of a high-salt diet. A recent study from Lanaspa et al. uncovered that high sodium consumption enhanced the aldose reductase-fructokinase pathway both in the liver and hypothalamus, promoting endogenous fructose generation and leptin resistance, and hyperphagia, which in turn led to obesity and obesity-induced NAFLD [8]. Additionally, such salt-induced NAFLD has been demonstrated by several clinical investigations, including both cross-sectional and prospective research studies [21,23,48,49,50,51]. The present study supplemented the previous literature and presented a profile of salt-induced lipid metabolism in female mice liver for the first time, providing a strong cue for salt use in modern diets and public health.

Our study has some limitations. First, the 8% dietary salt content in the present study was 20-fold higher than that fed to control mice (0.4%), which could not faithfully match the high-salt regime in daily life. Second, although the transcriptome results of our models are highly consistent with the phenotypes observed, this study still remains a descriptive study, and the precise molecular mechanisms need to be further investigated. To our knowledge, this study is for the first time to analyze the transcriptome throughout the livers of female mice fed a long-term high-salt diet. Thus, such a high-throughput-omics technology would bolster our ability to comprehensively investigate the complex disease progression and treatment response in metabolic disorders.

## 5. Conclusions

In summary, our study did a transcriptomic analysis in normal-diet and HSD female mice. We have identified quite a lot of DEGs, which are potentially associated with crucial metabolic processes such as glucose and lipid metabolism. Our findings might serve as a reference for further investigation of diet-induced metabolic disorders.

## Figures and Tables

**Figure 1 genes-14-00966-f001:**
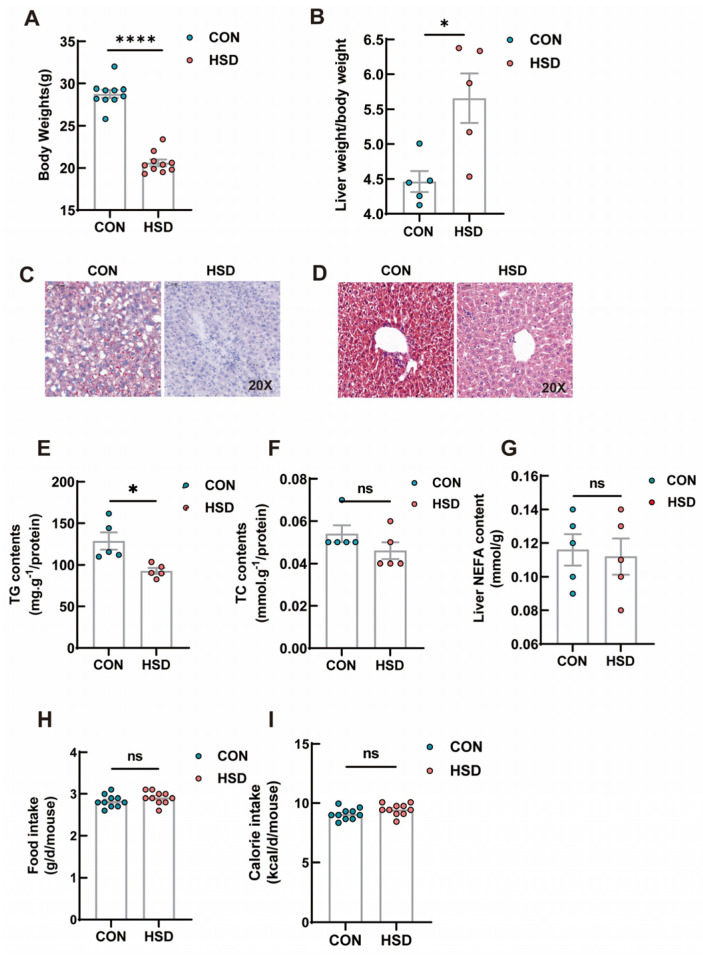
Phenotypic measurements of the effects of HSD on liver lipid metabolism. (**A**) Body weight of CON and HSD groups (*n* = 10). (**B**) Liver weight/body weight in CON and HSD groups (*n* = 5). (**C**) Representative H&E staining images of liver tissue after 12 weeks (scale bar = 50 μm). (**D**) Representative images of Oil Red O−stained livers (scale bar = 50 μm). (**E**–**G**) Total triglyceride (TG), cholesterol (TC) and non-esterified fatty acids (NEFA) levels in the livers after 12 weeks of feeding (*n* = 5). (**H**,**I**) Daily food intake(g/d/mouse) and calorie intake(kcal/d/mouse) of CON and HSD groups (*n* = 10). Data are presented as mean ± SEM. * *p* < 0.05; **** *p* < 0.0001, ns stands for non-significant difference (*p* > 0.05).

**Figure 2 genes-14-00966-f002:**
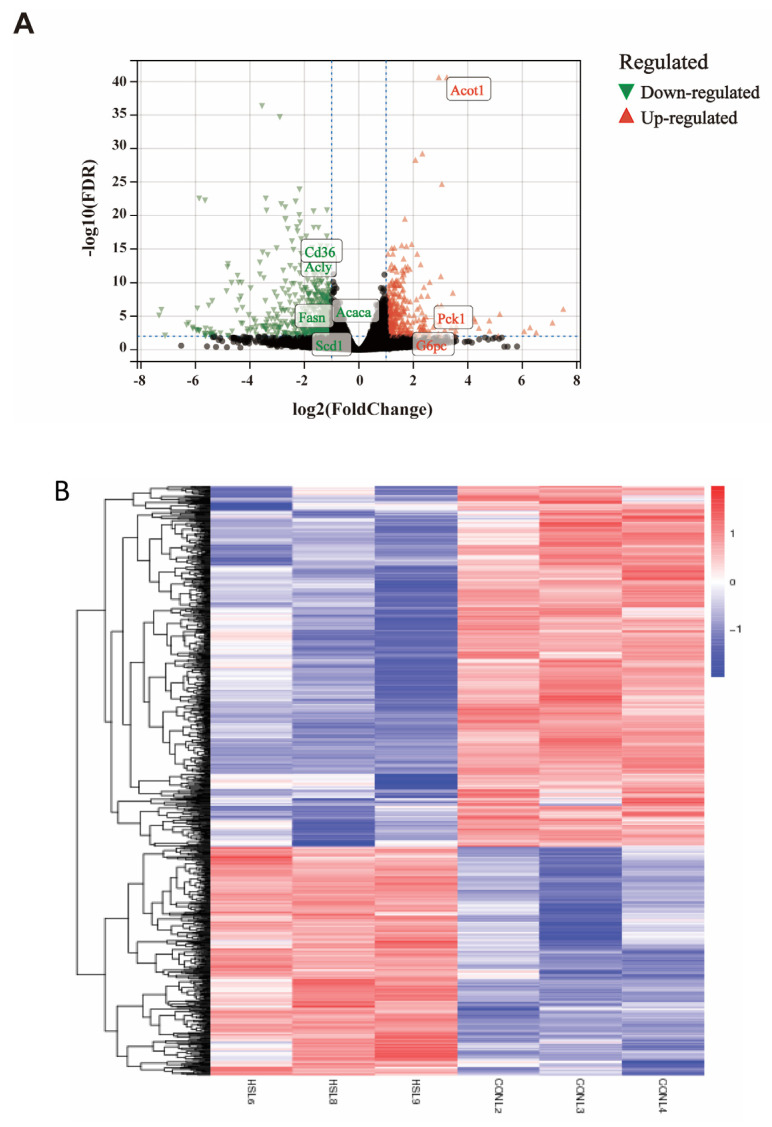
The volcano plots of DEGs of the two groups. (**A**) Volcano plot profiles−log10 Padj−value and log2−fold change of gene expression between CON vs. HSD liver samples (significantly altered defined as a Padj−value < 0.01). The green points indicate down−regulated genes, and the red points indicate up−regulated genes. (**B**) Heat map diagram of DEGs between CON and HSD mice.

**Figure 3 genes-14-00966-f003:**
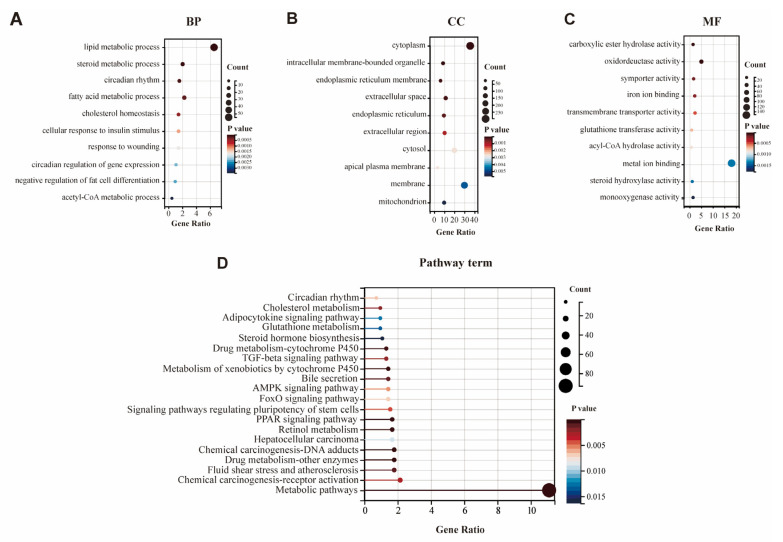
Significantly enriched GO terms and KEGG pathway enriched by differentially expressed genes after 12 weeks of HSD. Top 10 BP (**A**), CC (**B**) and MF (**C**) terms in the enrichment analysis of DEGs in the liver. (**D**) Top 20 KEGG pathways involved in liver metabolism enriched by differentially expressed genes.

**Figure 4 genes-14-00966-f004:**
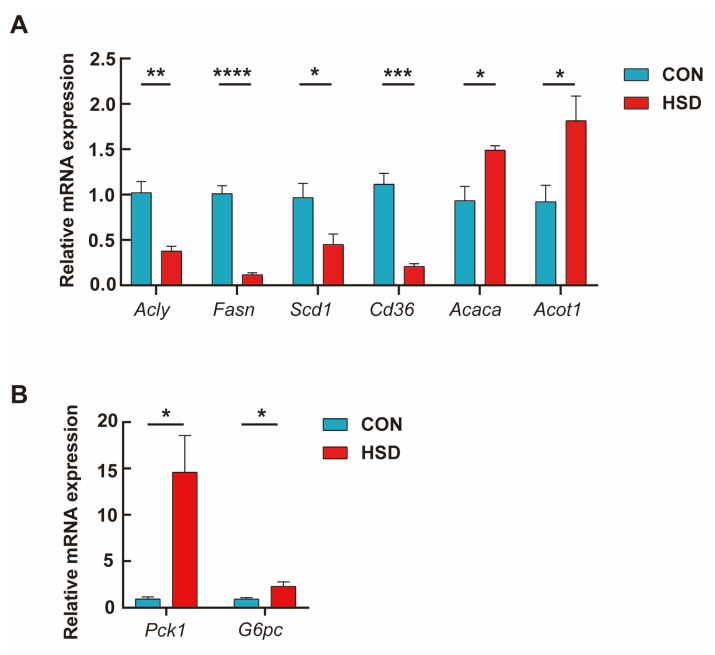
Quantitative real-time PCR analysis. (**A**) Results of six representative DEGs of the liver in lipid metabolism pathways (*Acly*, *Fasn*, *Scd1*, *Cd36*, *Acaca*, and *Acot1*). (**B**) Results of two representative DEGs of the liver in gluconeogenesis (*PCK1* and *G6PC*). Data represent means ± SEM. * *p* < 0.05; ***p* < 0.01; *** *p* < 0.001; **** *p* < 0.0001 (*n* = 3 biological replicates).

**Table 1 genes-14-00966-t001:** The 20 most downregulated genes in the high-salt diet mouse liver.

Gene Symbol	Gene Description	Log2fold-Change	Padj	Known Function
Down-Regulated				
Gm28548	Predicted gene 28548	−7.3357	7.94 × 10^−6^	LncRNA gene; unknown function
Gm40457	Predicted gene 40457	−7.25067	1.10 × 10^−6^	LncRNA gene; unknown function
Sult2a3	Sulfotransferase family 2A	−7.11578	0.008238	Hepatic sulfonation of bile acid [27]
Gm32468	Predicted gene 32468	−6.67475	5.61 × 10^−44^	LncRNA gene; unknown function
Olfr702	Olfactory receptor family 13 subfamily N member 4	−6.3241	0.000196	Upstream of or within G protein-coupled receptor signaling pathway and sensory perception of smell [28]
Igkv3-2	Immunoglobulin kappa variable 3-2	−6.2531	0.000105	Immune response [29]
Gm43626	Predicted gene 43626	−6.09587	0.000763	Pseudogene; unknown function
AC166052.1	Hypothetical protein I79_021887	−6.09463	0.000458	Unknown function
Olfr703	Olfactory receptor family 2 subfamily AG member 19	−5.96724	0.001053	G protein-coupled receptor signaling pathway [28]
D830013O20Rik	RIKEN cDNA D830013O20 gene	−5.94681	0.001425	LncRNA gene; unknown function
Pla2g4f	Phospholipase A2, group IVF	−5.93077	6.99 × 10^−7^	Protein coding gene; glycerophospholipid catabolic process [29]
5330426L24Rik	RIKEN cDNA 5330426L24 gene	−5.89239	0.001175	LncRNA gene; unknown function
Igkv15-103	Immunoglobulin kappa chain variable 15-103	−5.86438	0.001724	Immune response [29]
Cdh19	Cadherin 19, type 2	−5.86188	2.76 × 10^−23^	Adherens junction organization [29]
Psg18	Pregnancy-specific beta-1-glycoprotein 18	−5.85541	0.001422	Regulation of immune system process [29]; regulation of interleukin-10 production [30]
Mup-ps10	Major urinary protein, pseudogene 10	−5.65876	0.004965	Pseudogene; unknown function
Cap2	CAP, adenylate cyclase-associated protein, 2 (yeast)	−5.64668	5.44 × 10^−23^	cAMP-mediated signaling [29]
Selenok-ps7	Selenoprotein K, pseudogene 7	−5.63814	0.009325	Pseudogene; unknown function
6430710C18Rik	RIKEN cDNA 6430710C18 gene	−5.63211	0.00067	LncRNA gene; unknown function
Zfp385c	Zinc finger protein 385C	−5.62978	0.003875	Enables nucleic acid binding activity and zinc ion binding activity and is predicted to be active in the nucleus [29]

**Table 2 genes-14-00966-t002:** The 20 most upregulated genes in the high-salt diet mouse liver.

Gene Symbol	Gene Description	Log2fold-Change	Padj	Known Function
Up-Regulated				
Ttll8	Tubulin tyrosine ligase-like family, member 8	7.505617986	8.68 × 10^−7^	Flagellated sperm motility and protein polyglycylation [31]
Serpina4-ps1	Serine (or cysteine) peptidase inhibitor, clade A, member 4, pseudogene 1	7.110133615	8.81 × 10^−5^	Pseudogene; unknown function
Gm45301	Predicted gene 45301	6.512918923	0.0027869	LncRNA gene; unknown function
Ubap1l	Ubiquitin-associated protein 1-like	6.285266885	0.0004596	Ubiquitin-dependent protein catabolic process via the multivesicular body sorting pathway [29]
Scara5	Scavenger receptor class A, member 5	6.070547377	0.0017622	Enables ferritin receptor activity [32]
Plin4	Perilipin 4	5.174215259	4.81 × 10^−6^	Located in plasma membranes; unknown function
Gm8251	Predicted gene 8251	5.118764438	0.0088949	Protein coding gene; unknown function
Gm43305	Predicted gene 43305	4.796490714	0.0015396	LncRNA gene; unknown function
Slco1a1	Solute carrier organic anion transporter family, member 1a1	4.791203008	4.28 × 10^−5^	Enables organic anion transmembrane transporter activity; response to stilbenoid [33,34]
Cyp3a11	Cytochrome P450, family 3, subfamily a, polypeptide 11	4.71643503	4.06 × 10^−76^	Enables monooxygenase activity; upstream of or within the response to bacterium [35,36]
1700045H11Rik	RIKEN cDNA 1700045H11 gene	4.407589228	0.0005687	LncRNA gene; unknown function
Tmc7	Transmembrane channel-like gene family 7	4.275886302	5.53 × 10^−5^	Enable mechanosensitive ion channel activity [29]
Sftpa1	Surfactant-associated protein A1	4.228839366	2.17 × 10^−5^	Positive regulation of phagocytosis [29,37]
Dnaic1	Dynein axonemal intermediate chain 1	3.580195937	0.0003139	Enables both dynein heavy and light chain binding activity; insulin receptor signaling pathway [38]
Prkag3	Protein kinase, AMP-activated, gamma 3 non-catalytic subunit	3.553918438	0.0013324	Contributes to AMP-activated protein kinase activity; upstream of or within glycogen biosynthetic process; glycolytic process [29,39]
Cdkn1a	Cyclin-dependent kinase inhibitor 1A (P21)	3.525502285	0.0095903	Enables cyclin binding activity [29,40,41]
Omd	Osteomodulin	3.523000255	0.0063899	Regulation of bone mineralization; upstream of or within cell adhesion [29]
CT010575.2	Mus musculus chromosome 13 clone RP23-217J21	3.501322391	3.69 × 10^−5^	Long intervening noncoding RNAs (lincRNAs)
Fam222a	Family with sequence similarity 222, member A	3.450260655	3.20 × 10^−9^	Protein coding gene; unknown function
Serpina9	Serine (or cysteine) peptidase inhibitor, clade A (alpha-1 antiproteinase, antitrypsin), member 9	3.392001189	0.0004838	Enable serine-type endopeptidase inhibitor activity [29]

**Table 3 genes-14-00966-t003:** Differentially expressed genes in the liver involved in glucose and lipid metabolism in female mice fed with high salt diets (8% NaCl) for 12 weeks.

Genes	Annotation	Function	HSD vs. CON Log2fold-Change *
Pck1	Phosphoenolpyruvate carboxykinase 1	Gluconeogenesis	2.37
G6Pase	Glucose-6-phosphatase	Gluconeogenesis	1.97
Acss2	Acyl-CoA synthetase short-chain family member 2	Fatty acid synthesis	−2.70
Fasn	Fatty acid synthase	Fatty acid synthesis	−1.17
Acaca	Acetyl-CoA carboxylase alpha	Fatty acid synthesis	−1.02
Me1	Malic Enzyme 1	Fatty acid synthesis	−1.84
Scd1	Stearoyl-Coenzyme A desaturase 1	Fatty acid synthesis	−1.72
Elovl6	ELOVL fatty acid elongase 6	Fatty acid synthesis	−1.64
Acly	ATP citrate lyase	Fatty acid synthesis	−2.12
Acc	Acetyl-CoA carboxylase	Fatty acid synthesis	−1.26
Cd36	CD36 molecule	Fatty acid transporter	−2.12
Fabp5	Fatty acid binding protein 5	Fatty acid transporter	−2.27
Acsl3	Acyl-CoA synthetase long-chain family member 3	Fatty acid transporter	−0.92
Cyp4a10	Cytochrome P450, family 4, subfamily a, polypeptide 10	Fatty acid oxidation	1.65
Cyp4a14	Cytochrome P450, family 4, subfamily a, polypeptide 14	Fatty acid oxidation	1.44
Cyp4a31	Cytochrome P450, family 4, subfamily a, polypeptide 31	Fatty acid oxidation	2.53
Sc5d	Sterol-C5-desaturase	Cholesterol synthesis	−1.66
Gpam	Glycerol-3-phosphate acyltransferase	Triglyceride synthesis	−1.36
Cyp7a1	Cholesterol 7α hydroxylase	Bile acids synthesis	1.30
Cyp17a1	Cytochrome P450 family 17 subfamily A member 1	Steroidogenesis	−1.05

***** Log2 fold-change is expressed as the logarithmic fold change between the two groups. The positive Log2 fold-change indicates the upregulated genes, and the negative Log2 fold-change indicates downregulated genes.

**Table 4 genes-14-00966-t004:** Top 10 most significantly enriched canonical signaling pathways identified in liver samples of feed high-salt female mice.

Canonical Pathways	Map Number	−log(*p*-Value)	Gene
Metabolic pathways	map01100	1.06 × 10^−6^	*Colgalt2*, ***Alas1***, ***Rdh9***, ***Cyp3a11***, *Hnmt*, ***Lipg***, *Pde4b*, *Me1*, *Scd1*, ***Kmt5a***, *Nat8f7*, ***Glul***, *Pla2g4f*, ***G6pc***, ***Cyp4a10***, ***Cad***, *Cyp3a44*, *Acsl5*, *Sirt5*, ***Cyp4a14***, *Ctps*, *Gcat*, *Inmt*, *Acly*, *Dct*, *Cyp2a4*, *Ehhadh*, *Cyp3a41b*, ***Pgp***, *Adssl1*, *Acot2*, ***Acot1***, *Adprm*, *Aldh1a7*, ***Mgll***, ***Dgkh***, *Ndufb9*, ***Pfkfb3***, ***Nnmt***, *Galt*, *Acss2*, *Pde1a*, *Mgst3*, *Gstp1*, ***Hdc***, *Haao*, *Atp5k*, *Gstt2*, *Adcy1*, ***Acnat2***, ***Ak6***, ***Cyp7a1***, ***Acacb***, *Papss2*, *Acaca*, *Tymp*, *Cyp17a1*, *Neu2*, *Rdh11*, *Ugt1a5*, *Rdh16*, ***Atp6v0a2***, *Sc5d*, *Hyi*, *St3gal6*, *9130409i23rik*, ***Ugt1a9***, ***Pck1***, *Hao2*, *Gstm4*, *Car3*, *Nqo1*, *Gstm2*, *Car1*, *Cyp4a31*, *Sqor*, ***Ugt2b37***, ***Setd1a***, ***Dhcr24***, *Nmnat3*, *Mthfs*, *Fmo5*, *Qdpr*, *Rpia*, *Gale*, *Gpam*, *Gsta4*, *Gnpda2*, ***P4ha2***, *Gsta2*, *Fasn*, *Echdc1*, *Cryl1*
Chemical carcinogenesis-DNA adducts	map05204	1.30 × 10^−6^	*Gstm4*, *Gstm2*, *Mgst3*, *Gstp1*, ***Ugt2b37***, *Cyp3a44*, ***Cyp3a11***, *Gstt2*, *Gsta4*, *Gsta2*, *Cyp3a41b*, *Ugt1a5*, *Sult2a7*, ***Ugt1a9***, *Sult2a3*
Drug metabolism-other enzymes	map00983	4.00 × 10^−6^	*Gstm4*, *Gstm2*, *Mgst3*, *Gstp1*, ***Ugt2b37***, *Gstt2*, *Tymp*, *Ces2c*, *Gsta4*, *Ces2e*, *Ces1e*, *Gsta2*, *Ces2h*, *Ugt1a5*, ***Ugt1a9***
PPAR signaling pathway	map03320	1.40 × 10^−5^	***Cyp4a31***, ***Cyp4a10***, *Acsl5*, ***Cyp4a14***, ***Cyp7a1***, *Fabp5*, *Ehhadh*, ***Plin4***, *Me1*, *Plin2*, *Cd36*, *Scd1*, ***Pck1***, ***Plin5***
Retinol metabolism	map00830	3.60 × 10^−5^	***Rdh9***, ***Cyp4a31***, ***Cyp4a10***, ***Ugt2b37***, *Cyp3a44*, ***Cyp3a11***, ***Cyp4a14***, *Cyp2a4*, *Rdh11*, *Cyp3a41b*, *Rdh16*, *Ugt1a5*, ***Ugt1a9***, *Aldh1a7*
Metabolism of xenobiotics by cytochrome P450	map00980	4.80 × 10^−5^	*Gstm4*, *Gstm2*, *Gsta4*, *Gstp1*, *Gsta2*, *Mgst3*, ***Ugt2b37***, *Ugt1a5*, *Sult2a7*, *Gstt2*, ***Ugt1a9***, *Sult2a3*
Drug metabolism-cytochrome P450	map00982	1.89 × 10^−4^	*Gstm4*, *Gstm2*, *Gsta4*, *Gstp1*, *Gsta2*, *Mgst3*, ***Ugt2b37***, *Ugt1a5*, *Gstt2*, ***Ugt1a9***, ***Fmo5***
Fluid shear stress and atherosclerosis	map05418	7.26 × 10^−4^	*Gstm4*, *Nqo1*, *Gstm2*, ***Hsp90aa1***, ***Il1r1***, ***Dusp1***, *Itgb3*, *Mgst3*, *Gstp1*, *Gstt2*, *Acvr2b*, ***Thbd***, *Gsta4*, *Gsta2*, ***Rac3***
Bile secretion	map04976	8.14 × 10^−4^	*Abcg8*, ***Slco1a1***, *Aqp8*, ***Ugt2b37***, *Ugt1a5*, *Sult2a7*, *Adcy1*, ***Acnat2***, ***Ugt1a9***, *Abcb1a*, ***Cyp7a1***, *Sult2a3*
Cholesterol metabolism	map04979	0.001805528	***Mylip***, *Abcg8*, ***Sort1***, *Angptl8*, ***Lipg***, *Apoa4*, *Cd36*, ***Cyp7a1***

Up-regulated genes in high-salt mice are highlighted in bold and down-regulated genes in normal typeface.

## Data Availability

All the RNA-seq data are provided in Data S1 and data that support the findings of this study are available from the corresponding author upon reasonable request.

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
