# Peer review of "Hepatic Transcriptomics Reveals Reduced Lipogenesis in High-Salt Diet Mice"

_genes, 2023, doi:10.3390/genes14050966_

Round 1

Reviewer 1 Report

The manuscript entitled "Hepatic Transcriptomics Reveals Reduced Lipogenesis in High-Salt Diet Mice" contains significant discoveries on that a high-salt diet (HSD) increases the risk of cardiovascular disease and metabolic dysfunction. The title of the article is concise and informative, as it clearly conveys the main finding of the study. 

The introduction provides a good background and rationale for the study, as it highlights the health risks associated with a high-salt diet and the lack of knowledge about its impact on hepatic metabolism. However, it would be beneficial to provide more specific details about the research questions and hypotheses that the study aims to address. The methods section provides a clear and detailed description of the transcriptome analysis and RT-qPCR validation. 

The results section presents the key findings of the study, including the identification of down-regulated genes related to lipid and steroid biosynthesis in the livers of HSD mice. The presentation of the results is clear and organized, and the use of figures and tables helps to illustrate the findings.

Finally, the discussion section provides a thorough interpretation of the results and their implications for future research. The authors appropriately acknowledge the limitations of the study and suggest possible mechanisms underlying the observed changes in gene expression. However, it would be beneficial to include more discussion about the potential clinical relevance of the findings and their implications for public health.

Overall, the article provides a well-executed and informative study that sheds light on the molecular mechanisms underlying the impact of HSD on hepatic metabolism. With some improvements in title specificity and discussion of clinical relevance, the article could become a valuable resource for researchers and clinicians interested in the link between diet and metabolic health. However, the authors suggested to refer the latest citations to support the statement and reduce the number of similarities in the text.

Author Response

Dear Editors,

We are very grateful for the review comments provided by the editors and the external reviewers with regard to our manuscript entitled “Hepatic Transcriptomics Reveals Reduced Lipogenesis in High-Salt Diet Mice” (Manuscript ID: genes-2315584). We have endeavored to address all of the comments and suggestions and have revised the manuscript accordingly. All the changes in the manuscript are highlighted in the manuscript. Please see below our responses to the comments that the reviewers have raised.

Thanks and best regards,

Yao Li & Xiaoying Li

Specific comments:

 Reviewer 1

-1 The introduction provides a good background and rationale for the study, as it highlights the health risks associated with a high-salt diet and the lack of knowledge about its impact on hepatic metabolism. However, it would be beneficial to provide more specific details about the research questions and hypotheses that the study aims to address. The methods section provides a clear and detailed description of the transcriptome analysis and RT-qPCR validation.

Response:

We appreciated the professional suggestions by the reviewer. We have added and highlighted this part in the Introduction section (page 2, lines 56-58) in our revised manuscript. Thanks again to the reviewer for these professional suggestions.  

-2 Finally, the discussion section provides a thorough interpretation of the results and their implications for future research. The authors appropriately acknowledge the limitations of the study and suggest possible mechanisms underlying the observed changes in gene expression. However, it would be beneficial to include more discussion about the potential clinical relevance of the findings and their implications for public health.

Response:

We are extremely grateful for your guidance and help. We have added and highlighted this part into the Discussion section (page 14, lines 289-292) in our revised manuscript.

-3 Overall, the article provides a well-executed and informative study that sheds light on the molecular mechanisms underlying the impact of HSD on hepatic metabolism. With some improvements in title specificity and discussion of clinical relevance, the article could become a valuable resource for researchers and clinicians interested in the link between diet and metabolic health. However, the authors suggested to refer the latest citations to support the statement and reduce the number of similarities in the text.

Response:

Thanks for your nice comments. A new reference (Luo et al., 2022; doi:10.3389/fnut.2022.930316) was added to describe the effect of sodium intake in the liver (reference 36 in page 16, lines 398-400). Again, thanks for this point.

We look forward to hearing from you regarding our submission. We would be glad to respond to any further questions and comments that you may have.

Reviewer 2 Report

Re:'Hepatic Transcriptomics Reveals Reduced Lipogenesis in 2 High-Salt Diet Mice"

The paper is well written, I have some comments for the authors to address;

1- The total number of mice used were not disclosed however, after randomly assigning groups, you had extracted liver tissue from 3 mice on HSD and 3 mice on Con diets. Can you justify why and how you picked 3 liver tissues from each group? Also the number of mice in each group was showing10 from body weight graph however, only 5 samples for the biochemical assays? can you please explain.

2- How were the mice housed? in groups? or individually?

3- Your data shows that there is significant downregulation in lipogenic genes, would this data have been influenced by an isocaloric diet? since high salt intake can cause hyperphagia as found in one of the studies.

4- Is there water intake data?

5- Authors did not propose how this data may influence obesity, metabolic syndrome and insulin resistance.

6- was glucose and insulin resistance evident in these mice?

7- minor errors to be corrected: extra space to be removed 'line 15', rewording of last sentence 'line 47 and 48', add the word 'and to the start of 'line 53', extra space to be removed 'line 106', remove the word 'the' from 'line 232', reword sentence in 'line 263' to be more clear.

Author Response

Dear Editors,

We are very grateful for the review comments provided by the editors and the external reviewers with regard to our manuscript entitled “Hepatic Transcriptomics Reveals Reduced Lipogenesis in High-Salt Diet Mice” (Manuscript ID: genes-2315584). We have endeavored to address all of the comments and suggestions and have revised the manuscript accordingly. All the changes in the manuscript are highlighted in the manuscript. Please see below our responses to the comments that the reviewers have raised.

Thanks and best regards,

Yao Li & Xiaoying Li

Specific comments:

 Reviewer 2

1- The total number of mice used were not disclosed, however, after randomly assigning groups, you had extracted liver tissue from 3 mice on HSD and 3 mice on Con diets. Can you justify why and how you picked 3 liver tissues from each group? Also, the number of mice in each group was showing 10 from body weight graph, however, only 5 samples for the biochemical assays? can you please explain.

Response:

We really appreciate the reviewer for pointing out this issue. Mice were randomly divided into two groups with ten mice in each group. We are very sorry for not making it clear that each dot represents a pooled liver sample from two mice per group. Thank you for reading our manuscript carefully.

2- How were the mice housed? in groups? or individually?

Response:

Thank you for your valuable question. Mice were group-housed with 5 mice per cage, on a 12 h light–dark cycle.

3- Your data shows that there is significant downregulation in lipogenic genes, would this data have been influenced by an isocaloric diet? since high salt intake can cause hyperphagia as found in one of the studies.

Response:

Thank you very much for your valuable question. As previously described by Lanaspa et al., high salt intake indeed could lead to hyperphagia via increase in endogenous fructose production. Here, we managed to measure food intake and caloric intake in our study, and we presented this part of results in the manuscript (Please see Figure1 H-I). Briefly, both food intake and the caloric intake showed no significant difference between groups. In this sense, the downregulation in lipogenic genes can be thought of an outcome in response to an isocaloric state.

4- Is there water intake data?

Response:

Thank you for your valuable question. Mice had unrestricted access to food and water through the course of the study. We managed to measure water consumption in our study, and found that the 24-hour water intake was significantly higher in the high-salt diet mice than in the control group, which was the effect of a high-salt regime on mice model.

5- Authors did not propose how this data may influence obesity, metabolic syndrome and insulin resistance.

Response:

Thank you very much for your valuable question. As we have stated in the Introduction section, elevated sodium intake is related to the development of many metabolic disturbances, including obesity, metabolic syndrome and insulin resistance. Besides, we have also reviewed the potential mechanisms for the relationship between salt consumption and Non-alcoholic fatty liver disease (NAFLD): elevation of caloric consumption, hyperosmolarity induced by salt, insulin resistance, endogenous fructose synthesis, and dysfunction of the RAAS (Xu J et al., 2022; doi:10.1038/s41430-021-01044-8). In contrast to previous findings, a significant decrease in lipid deposition was observed in this study, and this result may have been due to the higher concentration of salt intake in our experiment. So, we felt very sorry that we cannot make any further deductions between high salt intake (8% Nacl) and metabolic disorders such as obesity metabolic syndrome and insulin resistance you have mentioned, at this time. Nevertheless, it is still a critical topic worthy to be further investigated. Thank you again for your good question and nice comments.

6- Was glucose and insulin resistance evident in these mice?

Response:

Thanks for our reviewers for raising such a key question. During this experiment, we managed to perform both the random and fasting glucose levels in both groups of mice, however, no differences were found for random and fasting glucose concentrations. Hence, such a high salt diet appears to have weak effects on glucose homeostasis in mice. Additionally, with respect to insulin resistance, the central link of various metabolic abnormalities (Petersen MC et al., 2018; doi:10.1152/physrev.00063.2017), a further insulin resistance experiment was not conducted due to the weak impact in blood glucose. However, according to the previous literature, a high salt diet may promote insulin resistance in rats (Ogihara T et al., 2001; doi: 10.2337/diabetes.50.3.573), and this remains to be clarified in the future.

7- minor errors to be corrected: extra space to be removed 'line 15', rewording of last sentence 'line 47 and 48', add the word 'and to the start of 'line 53', extra space to be removed 'line 106', remove the word 'the' from 'line 232', reword sentence in 'line 263' to be more clear.

Response:

We gratefully thank the reviewer for pointing out this problem. We have corrected all of the above. Thank you for reading our manuscript carefully!

We look forward to hearing from you regarding our submission. We would be glad to respond to any further questions and comments that you may have.

Reviewer 3 Report

Reviewer's comments:

In this study, the authors investigated how intake of a high-salt diet (HSD) affects gene expression in the liver. They found that genes involved in fatty acid synthesis were down-regulated, whereas genes related to fatty acid oxidation were significantly up-regulated in response to high-salt stimulation. The authors concluded that the detected differences in gene expression may contribute to revealing the effect of HSD on lipid and glucose metabolism in the liver.

It is necessary to clarify several facts:

1) The availability of laboratory technical equipment enables detailed analysis of gene expression, but unfortunately the phenotypic conditions are forgotten.

2) For the interpretation of the gene analysis in the liver, it is necessary to indicate the nutritional state in which the tissue was collected. It is understood that gene regulation is affected by fasting or postprandial state. Add this data to the methodological part.

3) Animal models should be as close as possible to conditions in humans. Why did the authors use such an enormously high salt content in the diet (8%) and in addition added NaCl (0.9%) to the drinking water? After all, even 1% salt content in the diet affects, for example, blood pressure in animals. 4% NaCl concentration in the diet is considered as the maximum content leading to metabolic disorders. Clarify on the basis of what knowledge you have chosen the salt content in the diet.

4) Extraction of TAG, cholesterol and NEFA from tissues before their determination with the kit is quite laborious. Please describe the extraction method and determination principle in the methodological section.

5) In Fig. 1E and 1F , Tg and cholesterol concentrations in the liver are given per protein, while NEFA concentrations in Figs. 1G are expressed per tissue weight. You should explain a) why you used the conversion to protein, which is not the usual way of expression, nor does it have a physiological rationale;  b) whether the concentration of proteins changed due to HSD; c) state in the methodological section which method was used to analyse the proteins in the liver.

6) The authors discuss studies showing a possible effect of HSD on adipocyte hypertrophy. Could the authors add their data on visceral amounts of fat bodies? These data are important because there is a constant circulation of lipids between the adipose tissue and the liver.

7) Most animal studies show that HSD increases hepatic steatosis and Tg accumulation, explain the difference with your study, where genes involved in lipid synthesis and transport were down-regulated and Tg concentration was reduced.

8) The methods section states that results are expressed as mean ± SD, but in the results they are reported as mean ± SEM.

Author Response

Dear Editors,

We are very grateful for the review comments provided by the editors and the external reviewers with regard to our manuscript entitled “Hepatic Transcriptomics Reveals Reduced Lipogenesis in High-Salt Diet Mice” (Manuscript ID: genes-2315584). We have endeavored to address all of the comments and suggestions and have revised the manuscript accordingly. All the changes in the manuscript are highlighted in the manuscript. Please see below our responses to the comments that the reviewers have raised.

Thanks and best regards,

Yao Li & Xiaoying Li

Specific comments:

 Reviewer 3

In this study, the authors investigated how intake of a high-salt diet (HSD) affects gene expression in the liver. They found that genes involved in fatty acid synthesis were down-regulated, whereas genes related to fatty acid oxidation were significantly up-regulated in response to high-salt stimulation. The authors concluded that the detected differences in gene expression may contribute to revealing the effect of HSD on lipid and glucose metabolism in the liver.

It is necessary to clarify several facts:

1) The availability of laboratory technical equipment enables detailed analysis of gene expression, but unfortunately the phenotypic conditions are forgotten.

Response:

We really appreciate the reviewer for pointing out this issue. Although the RNA-seq is the more sensitive technique, the detect of the phenotype is of crucial importance to discover the precise association signals. And the content of phenotypic test is relatively deficient in our study. Therefore, other HSD-induced phenotypes remain to be examined in the future study.

2) For the interpretation of the gene analysis in the liver, it is necessary to indicate the nutritional state in which the tissue was collected. It is understood that gene regulation is affected by fasting or postprandial state. Add this data to the methodological part.

Response:

We appreciated the professional suggestions by the reviewer. To measure liver biochemical markers, the mice were sacrificed after 6 hours of fasting. And all the livers were harvested and snap frozen in liquid N2. We have added and highlighted this part in the methodological section (page 2, Lines 78-79).

3) Animal models should be as close as possible to conditions in humans. Why did the authors use such an enormously high salt content in the diet (8%) and in addition added NaCl (0.9%) to the drinking water? After all, even 1% salt content in the diet affects, for example, blood pressure in animals. 4% NaCl concentration in the diet is considered as the maximum content leading to metabolic disorders. Clarify on the basis of what knowledge you have chosen the salt content in the diet.

Response:

We are appreciative of the reviewer’s suggestion. According to previous reports, such a much higher increment in salt administration (4%-8%) was frequently used to investigate the cardiovascular effects of a high-sodium diet (Simon G. 2003; doi: 10.1016/j.amjhyper.2003.07.019; De Miguel C et al., 2019, doi: 10.1111/apha.13227; Hu L et al., 2020, doi: 10.3233/JAD-200035). It is at this point that 8% Nacl concentration was chosen for investigation in our current study. And we have to acknowledge such a much higher daily salt intake by mice (8% NaCl) may not be physiologic. Indeed, it has been estimated, based on body weights and metabolic rates, that a 2% NaCl diet of rats corresponds to a daily intake of 10 to 15 g of salt in human subjects (Folkow B.1992, doi:10.3109/10641969209036167). Thus, experiments in which physiologically relevant concentration of salt administered for prolonged periods of time indeed trump in regard to the importance of experiments with extremely high concentration of salt for a short period of time. Thank you again for your professional question and nice comments.

4) Extraction of TAG, cholesterol and NEFA from tissues before their determination with the kit is quite laborious. Please describe the extraction method and determination principle in the methodological section.

Response:

We really appreciate the reviewer for pointing out this issue. Hepatic lipids were extracted with chloroform methanol (2:1) as previously described (Xiao Y et al., 2016; doi: 10.2337/db15-1024). We added more details about the details in the methodological section (page 2, lines 80-81), adding a new reference (please see ref-26).

5) In Fig. 1E and 1F, Tg and cholesterol concentrations in the liver are given per protein, while NEFA concentrations in Figs. 1G are expressed per tissue weight. You should explain a) why you used the conversion to protein, which is not the usual way of expression, nor does it have a physiological rationale; b) whether the concentration of proteins changed due to HSD; c) state in the methodological section which method was used to analyse the proteins in the liver.

Response:

Thank you for your valuable question. Indeed, the formation of protein complexes is determined by a number of factors: physical properties of the protein itself, temperature, and concentration of the used cosolvent (salt). Solutes are frequently classified as chaotropes ('disorder-making'), which destabilize protein structures, and kosmotropes ('order-making,' which stabilize them) (Zangi R, doi: 10.1021/jp909034c; Beauchamp DL et al., doi: 10.1016/j.bpc.2011.11.004). Chaotropic salts interfere with intramolecular interactions mediated by non-covalent forces such as hydrogen bonds, van der Waals forces, and hydrophobic interactions, which, at high cosolvent concentrations, results in protein denaturation. Hence, salt could affect protein stability in a variety of ways. Therefore, hepatic triglyceride and total cholesterol were both quantified as mg/g protein. And we have added the details in described in the methodological section (page 2, lines 80-81). Also, this type of presentation for hepatic lipids can also be found in various research (Jump DB et al., doi: 10.1016/j.bcp.2010.12.014; Li X et al., doi: 10.1038/srep16774; Lytle KA et al., doi:10.1371/journal.pone.0146942). We sincerely hope that it is now better understandable. Thanks again for your insightful comment.

6) The authors discuss studies showing a possible effect of HSD on adipocyte hypertrophy. Could the authors add their data on visceral amounts of fat bodies? These data are important because there is a constant circulation of lipids between the adipose tissue and the liver.

Response:

We thank the reviewer for the nice question. Indeed, indicators of visceral amounts of fat bodies were not measured in this study. However, as suggested by our previous study (Mao et al., 2022. doi:10.3389/fendo.2022.887843), fat deposition in liver and WAT were found both decreased following ad lib feeding of the HSD diet for 12 weeks, which was not consistent with findings from studies in animals and human subjects (Lanaspa MA et al., 2018. doi: 10.1073/pnas.1713837115; DeClercq VC et al., 2016. doi: 10.3945/jn.115.227496). And we assumed that the discordance between our findings with previous studies might be attributed to difference in salt concentration.

7) Most animal studies show that HSD increases hepatic steatosis and Tg accumulation, explain the difference with your study, where genes involved in lipid synthesis and transport were down-regulated and Tg concentration was reduced.

Response:

We appreciate for your nice questions. This is also what we want to figure out in the experiment. As is well known, hepatic lipid deposition is strictly controlled by a variety of factors including dietary lipid intake, circulating lipid levels, glucose uptake, lipid synthesis, hepatic lipid oxidation, and liver lipid release. In our RNA-seq results of liver, we found genes related to lipogenesis were significantly decreased in mice of HSD group, including those involved in fatty acid and lipid biosynthesis and transport (Fasn, Scd1, Acly, Cd36 etc.), as well as the triglyceride synthesis (Gpam). And the results of RNA-seq in our high-salt mouse model are in well accordance with phenotypes observed in the liver. Therefore, we can see that the decreased lipid deposition in the liver of HSD is mainly attributed to the alteration of these genes involved in lipogenic process which eventually led to the decrease in lipid content. Nevertheless, it is still a critical topic worthy to be further investigated.

8) The methods section states that results are expressed as mean ± SD, but in the results, they are reported as mean ± SEM.

Thank you for reading our manuscript carefully. It has been corrected in the methodological section (page 3, lines 126-127), stated as “Data is expressed as mean ± standard mean of error (SEM)”. Again, we are so sorry for this carelessness.

We look forward to hearing from you regarding our submission. We would be glad to respond to any further questions and comments that you may have.

Round 2

Reviewer 3 Report

The authors addressed all my comments and several improvements have been made to the manuscript. However, not all of my previous comments were adequately addressed, which the authors say was due to the fact that the appropriate biological material was not collected during the experiment. The authors stated that my comments will help them in further experiments. The manuscript is now suitable for publication.